# RNA-Seq Provides Insights into the Mechanisms Underlying *Ilyonectria robusta* Responding to Secondary Metabolites of *Bacillus methylotrophicus* NJ13

**DOI:** 10.3390/jof8080779

**Published:** 2022-07-26

**Authors:** Xiang Li, Mengtao Li, Xiangkai Liu, Yilin Jiang, Dongfang Zhao, Jie Gao, Zhenhui Wang, Yun Jiang, Changqing Chen

**Affiliations:** 1College of Life Science, Jilin Agricultural University, Changchun 130118, China; xli88888@126.com; 2College of Plant Protection, Jilin Agricultural University, Changchun 130118, China; lmt19980722@163.com (M.L.); xiangkailiu8733@163.com (X.L.); jiegao115@163.com (Y.J.); jiegao115@126.com (J.G.); 3Jilin Provincial Agro-Tech Extension Center, Changchun 130031, China; mayflowers503@126.com; 4College of Agronomy, Jilin Agricultural University, Changchun 130118, China; wzhjlau@163.com

**Keywords:** *Bacillus methylotrophicus*, secondary metabolites, *Ilyonectria robusta*, transcriptome, differentially expressed fungal genes

## Abstract

(1) Background: *Ilyonectria robusta* can cause ginseng to suffer from rusty root rot. Secondary metabolites (SMs) produced by *Bacillus methylotrophicus* NJ13 can inhibit the mycelial growth of *I. robusta*. However, the molecular mechanism of the inhibition and response remains unclear. (2) Methods: Through an in vitro trial, the effect of *B. methylotrophicus* NJ13’s SMs on the hyphae and conidia of *I. robusta* was determined. The change in the physiological function of *I. robusta* was evaluated in response to NJ13’s SMs by measuring the electrical conductivity, malondialdehyde (MDA) content, and glucose content. The molecular interaction mechanism of *I. robusta*’s response to NJ13’s SMs was analyzed by using transcriptome sequencing. (3) Results: NJ13’s SMs exhibited antifungal activity against *I. robusta*: namely, the hyphae swelled and branched abnormally, and their inclusions leaked out due to changes in the cell membrane permeability and the peroxidation level; the EC_50_ value was 1.21% (*v*/*v*). In transcripts at 4 dpi and 7 dpi, the number of differentially expressed genes (DEGs) (|log_2_(fold change)| > 1, *p* adj ≤ 0.05) was 1960 and 354, respectively. NJ13’s SMs affected the glucose metabolism pathway, and the sugar-transporter-related genes were downregulated, which are utilized by *I. robusta* for energy production. The cell wall structure of *I. robusta* was disrupted, and chitin-synthase-related genes were downregulated. (4) Conclusions: A new dataset of functional responses of the ginseng pathogenic fungus *I. robusta* was obtained. The results will benefit the development of targeted biological fungicides for *I. robusta* and the study of the molecular mechanisms of interaction between biocontrol bacteria and phytopathogenic fungi.

## 1. Introduction

Ginseng (*Panax ginseng*), a plant with great medicinal value, has been used in medicinal research [1]. Ginseng can produce bioactive metabolites and has numerous pharmacological properties [2]. In recent years, ginseng has been widely cultivated with increasing productivity levels. However, *Ilyonectria robusta*, *Botrytis cinerea*, *Pythium* spp., and *Rhizoctonia solani* can cause ginseng diseases [3,4,5]. Among these pathogens, *I. robusta* is a soil-borne pathogenic fungus that is also destructive to ginseng. It has been detected in the rusty root rot of *P. ginseng* and is considered to be the main cause of ginseng rust root rot formation [5]. In addition to causing disease in ginseng, *Ilyonectria* spp. can cause black foot rot of grapevine [6,7], apple replant disease, and beech canker [8,9]. In Chongqing, China, *I. robusta* has been discovered for the first time to cause root rot in *Codonopsis tangshen* [10].

For a long time, the prevention and control of ginseng rust rot mainly adopted chemical control methods. However, the long-term use of chemical fungicides will lead to serious problems, such as the emergence of pathogenic bacterial resistance, a decline in the control effect, pollution of the environment, and an increase in pesticide residues [11]. Therefore, it is necessary to explore biological control agents (BCAs) and secondary metabolites (SMs) produced by fungi, bacteria mainly including *Bacillus* spp., and actinomycetes as effective eco-friendly alternatives to chemical fungicides [12]. In recent years, *Bacillus* spp. have been widely used as BCAs to control plant diseases [13] and are considered to be a major source of some secondary metabolically active substances [14]. The study of SMs is one of the important mechanisms of biological control and a research hotspot, which has both theoretical innovation and application potential. *Bacillus* spp. can synthesize cyclic lipopeptides (CLPs), which have broad-spectrum antibacterial activity [15]. For example, culture filtrates of *B. methylotrophicus* strain BC79 can control rice blast in the greenhouse and the field [16]. In addition, the crude bacteriocin substance of *B. methylotrophicus* BM47 can antagonize *Fusarium moniliforme*, *Aspergillus awamori*, and *Penicillium* sp. [17]. Difficidin and oxydifficidin are produced by *B. methylotrophicus* DR-08 and exhibit strong antibacterial activity against *Ralstonia solanacearum*, which causes tomato bacterial wilt [18]. These findings laid the foundation for the subsequent study of the response mechanisms of phytopathogenic fungi to secondary metabolites from biocontrol strains.

In the existing research on the biocontrol mechanisms of biocontrol SMs, there are many studies that focus on the physiological and biochemical levels. Fengycins were found to induce cell death in the cytoplasm and cell wall of fungal pathogenic hyphae by increasing cell permeability [19]. It is well-known that the accumulation of reactive oxygen species (ROS) in living organisms can oxidize various biological macromolecules in cells, leading to cell membrane damage or cell death [20]. Furthermore, among CLPs, in addition to fengycins, bacillomycin D, and iturins can also interact antagonistically by inducing ROS production in phytopathogenic fungi [21,22]. *Bacillus* spp. can produce non-ribosomal oligopeptide derivatives that disrupt the cellular structure of phytopathogenic fungi, which bind to the cytoplasmic membrane by targeting [23]. Crude metabolites of *Streptomyces corchorusii* can also cause cell membrane damage through defects in ergosterol synthesis and the accumulation of MDA in *F. oxysporum* f. sp. *niveum* [24]. The siderophore bacillibactin produced by *Bacillus* spp. can acquire Fe^3+^ and other metal ions in the environment, thereby competing with plant pathogens for metal elements required for growth [25,26]. High-throughput sequencing technologies (RNA-Seq) can be used to analyze the gene expression of most organisms under different conditions [27,28]. Based on RNA-Seq, the molecular mechanisms by which phytopathogenic fungi respond to biological control have been discussed [29].

The results of previous studies in which an endophytic *B. methylotrophicus* NJ13 isolated from the ginseng stem was chosen, which exhibited broad-spectrum antagonism to pathogenic fungi including *F. solani* and *Alternaria panax* in ginseng, showed a better control effect on Alternaria leaf spot of ginseng in the field [30,31,32]. At the same time, secondary metabolite gene clusters of strain NJ13 were predicted by genomic sequencing, with the results showing its potential to negatively affect pathogenicity of *I. robusta* CBLJ-3 [33]. Furthermore, an antifungal protein, FlgG, that had antagonistic activity against *I. robusta* was identified, and it was proven that the production of secondary metabolic antibiotics was one of the main biocontrol mechanisms for NJ13 strain [34].

The objectives of this study were to determine the inhibitory effect of *B. methylotrophicus* NJ13’s SMs, which has antagonistic activity against *I. robusta*, and to explore the types and expression levels of differentially expressed genes in order to analyze the response mechanism of *I. robusta* stressed by *B. methylotrophicus* NJ13’s SMs using RNA-Seq and genomic methods. The study is expected to lay the foundation for clarifying the mechanisms of antagonistic bacteria that control ginseng rusty root rot and for developing biological fungicides based on target genes of *I. robusta*. 

## 2. Materials and Methods

### 2.1. Isolation and Culture Conditions of Strains

*B. methylotrophicus* NJ13 and *I. robusta* CBLJ-3 were provided by the Laboratory of Integrated Management of Plant Diseases, Jilin Agricultural University, Changchun, Jilin Province, China, and stored at −80 °C. Strain CBLJ-3 was incubated at 25 °C for 3 d on potato dextrose agar (PDA) medium. Strain NJ13 was cultured at 28 °C for 2 d on Luria–Bertani (LB) agar medium.

### 2.2. Preparation of CBLJ-3’s Conidia and Fermentation Filtrates without NJ13’s Cells

A hyphal disk of strain CBLJ-3 with a diameter of 8 mm was inoculated into potato dextrose broth (PDB) liquid medium and incubated at 25 °C, 150 rpm, for 7 d. Later, the spore suspension was obtained through filtering by using eight layers of sterile gauze. The concentration of CBLJ-3 conidia was adjusted to 10^6^ spores/mL. A agar disc of the strain NJ13 with a diameter of 8 mm was firstly inoculated into the LB broth overnight and then was transferred, at inoculum volume of 10% (*v*/*v*), to 250 mL shake flasks with a volume of 100 mL and incubated in a new fermentation medium, which consisted of glucose (30.0 g/L), soluble starch (15.0 g/L), yeast extract (15.0 g/L), K_2_HPO_4_ (1.0 g/L), NaCl (5.0 g/L), and distilled water with a pH of 7.0; it was then cultured at 28 °C, 160 rpm, for 72 h. The fermentation broth was centrifuged at 8000 rpm for 20 min, and then, the fermentation filtrates without NJ13’s cells, named “NJ13 sterile filtrates”, were obtained through a 0.22 μm filter and were stored at 4 °C for use in the next day. 

### 2.3. Antifungal Activity Assay and Microscopic Observation

NJ13 sterile filtrates were added to PDA medium to make an agar plate with a final concentration of 0.5%, 1%, 2.5%, 5%, or 10% (*v*/*v*). A mycelial agar plug (8 mm diameter) of strain CBLJ-3 cultured for 7 d was incubated on the center of the PDA agar plate (9 cm diameter), and a PDA agar plate without NJ13 sterile filtrates was used as the control. All plates were cultured at 25 °C, and the colony diameter of strain CBLJ-3 was measured after 4 d. Inhibition rate (%) = 100 × (C − T)/C, where C represents the radial growth of the control (mm), and T represents the radial growth of the test culture (mm). Based on the linear regression of the colony diameter versus the log-transformed concentration of NJ13 sterile filtrates, the EC_50_ value for strain CBLJ-3 was determined [35]. All experiments were repeated three times. Images of CBLJ-3’s hyphae were captured using a Zeiss microscope Axio Imager 2.0 (Carl Zeiss, Heidelberg, Germany).

### 2.4. Effect of NJ13’s SMs on the Germination of I. robusta Conidia 

The conidial suspension (100 μL, 10^6^ spores/mL) of CBLJ-3 was incubated onto water agar (15 g/L agar, Solarbio Life Sciences, Beijing, China) with 0%, 10%, 20%, 40%, 50%, and 70% (*v*/*v*) concentrations of NJ13’s SMs and cultured for 9 h at 25 °C in the dark. Petri dishes were observed under visible light using a Zeiss microscope Axio Imager 2.0 (Carl Zeiss, Heidelberg, Germany). The germination percentage was calculated as the number of germinated spores/total spores × 100%. The experiment was performed according to a randomized complete block design with three replicates. Images of CBLJ-3’s conidia were captured using the abovementioned microscope. 

### 2.5. Measurement of Mycelial Extracellular Conductivity

A total of 100 μL of the above-described spore suspension was added to 50 mL of PDB medium at 28 °C after a 2-day incubation. After being centrifuged (Eppendorf Centrifuge 5810R) at 8000× *g* for 2 min and washed, it was resuspended the sediment in other 50 mL of water supplemented with different concentrations of NJ13 sterile filtrates (0%, 10%, 20%, 40%, 50%, and 70% (*v*/*v*)). The mixture was centrifuged (Eppendorf Centrifuge 5810R) at 12,000× *g* for 2 min after being co-incubated for 0 h, 12 h, 24 h, 36 h, 48 h, and 64 h at 25 °C. All experiments were performed in triplicate. The supernatant was used to determine the extracellular conductivity of strain CBLJ-3’s cells using a conductivity meter (INESA, Shanghai, China). 

### 2.6. Detection of Malondialdehyde (MDA) in CBLJ-3’s Mycelia in the Presence of NJ13’s SMs

A 1.0 mL conidial suspension of CBLJ-3 (1 × 10^6^ spores/mL) was inoculated into 100 mL PDB medium for 24 h at 25 °C, 150 rpm. Various concentrations (EC_50_, 5%, 10%, and 20% (*v*/*v*)) of the NJ13 sterile filtrates were added to the PDB medium and set as a blank control. After being cultured for 24 h, they were filtered through three layers of filter paper. For each treatment, 1 g of mycelium was washed with 0.05 mol/L PBS buffer (pH 7.4) 3 times. Liquid nitrogen was added to the wet mycelium of each treatment, and the mycelium was ground to a powder. Then, 0.05 mol/L PBS buffer (pH 7.4) was added to prepare bacterial tissue homogenate, which was centrifuged at 4 °C and 8000 rpm for 15 min; the supernatant was taken as the extract for the determination of MDA, which was performed spectrophotometrically using a MDA assay kit (Nanjing Jiancheng Bioengineering Institute, Nanjing, China). The experiments were repeated three times.

### 2.7. Measurement of the Glucose Produced by CBLJ-3 in the Presence of NJ13’s SMs 

A 1.0 mL conidial suspension of CBLJ-3 (1 × 10^6^ spores/mL) was added to 100 mL PDB medium, and then, the NJ13 sterile filtrates were added so that the concentration reached the EC_50_ value; a blank control was then set. The treatments were as follows: (i) strain CBLJ-3 and NJ13’s SMs, named “FS”; (ii) control 1: strain CBLJ-3 and water, named “FW”; (iii) control 2: PDB medium and water, named “MW”; (iv) control 3: PDB medium and strain NJ13’s sterile filtrates, named “MS”. All treatments were cultured at 25 °C, 150 rpm. The 6 mL liquid was sampled at 0, 12, 24, 36, 48, and 60 hpi and was centrifuged at 4 °C and 12,000 rpm for 20 min. The 3,5-dinitro-salicylic (DNS) method was used to determine the glucose content [36]. The experiment was repeated three times.

### 2.8. Control Effect on Ginseng Rusty Roots Infected by I. robusta Using NJ13’s SMs

Three-year-old detached ginseng roots were chosen for the in vivo experiment. Then, healthy roots were detached one day prior to inoculation and were washed, their surface disinfested with 75% alcohol, and rinsed with sterile water. Wounds with a 3 mm depth in each ginseng taproot were made by using a sterile pipette tip with a 2 mm diameter. In this assay, 10 μL of the NJ13 sterile filtrates (100% (*v*/*v*)) were added to each wound (each ginseng root has 3 to 5 wounds) in ginseng roots by a pipette. Fungal plugs of CBLJ-3 with a diameter of 8 mm were cut at 1 cm from the edge of the colony incubated at 25 °C for 7 d on PDA medium and were fastened to the surface of ginseng roots by sterile pipette tips. The inoculation trial design included five treatments: (i) the treatment with sterile water as the blank control; (ii) strain CBLJ-3’s hyphal plugs inoculated after one day of inoculation with sterile water as the first positive control; (iii) sterile water that was applied after one day of inoculation with CBLJ-3’s hyphal plugs as the second positive control; (iv) strain NJ13 sterile filtrates were inoculated after one day of inoculation with strain CBLJ-3’s hyphal plugs as the protective group; and (v) strain CBLJ-3’s hyphal plugs that were inoculated after one day of inoculation with strain NJ13 sterile filtrates as the curative group. One day later, all of strain CBLJ-3’s hyphal plugs were removed. All inoculation samples were placed in sterile Petri dishes with a diameter of 25 cm and incubated at 25 °C. Each root was examined for infection symptoms one day after infection, with further examinations performed every 2 days over a 9-day period. All experiments were repeated three times.

### 2.9. RNA Extraction and Sequencing

Strain CBLJ-3 was incubated in PDA medium for 4 days. CBLJ-3’s mycelia were treated with NJ13 sterile filtrates for 4 dpi and 7 dpi at EC_50_ (process is shown in Appendix A). Petri dishes without NJ13 treatment were used as a control. The mycelia were harvested and frozen in liquid nitrogen for RNA-Seq (Control: R4d and R7d, Treatment: BF_R4d and BF_R7d). After RNA extraction with TRIzol reagent (Invitrogen, Shanghai, China), the samples were placed on 1% agarose gels and monitored for RNA degradation and contamination. RNA integrity was assessed using the RNA Nano 6000 Assay Kit of the Bioanalyzer 2100 system (Agilent Technologies, Palo Alto, CA, USA).

A total of 1 μg RNA per sample was used as input material for the RNA sample preparations. The clustering of the index-coded samples was performed on a cBot Cluster Generation System using a TruSeq PE Cluster Kit v3-cBot-HS (Illumina, San Diego, CA, USA) according to the manufacturer’s instructions. After cluster generation, the library preparations were sequenced on an Illumina Novaseq platform, and 150 bp paired-end reads were generated.

### 2.10. Data Analysis

Raw data (raw reads) in FASTQ format were first processed through in-house Perl scripts. In this step, clean data (clean reads) were obtained by removing reads containing adapters, reads containing poly-N, and low-quality reads from raw data. At the same time, the Q20 and Q30 values and the GC contents of the clean data were calculated. All downstream analyses were based on clean data with high quality. The clean data for each sample were mapped to strain CBLJ-3’s reference genome (data not released) using Bowtie 2 (version 2.2.3) (Fast gapped: read alignment with Bowtie 2. Nature methods) from Langmead et al. [37]. Blastp (2.4.0, E-value < 10^−5^) was used to align all genes with the Nr, KOG, and KEGG databases for functional annotation. The GO (gene ontology) annotation of the genes was obtained using Blast2GO (version 4.1.9, Valencia, Spain) from Conesa et al. [38].

Differential expression analysis of two conditions/groups (two biological replicates per condition) was performed using the DESeq2 R package (version 1.20.0, Berlin, Germany) from Love et al. [39]. Gene ontology (GO) enrichment analysis of differentially expressed genes was implemented using the clusterProfiler R package in which gene length bias was corrected [40]. GO terms with corrected *p*-values less than 0.05 were considered significantly enriched by differentially expressed genes. Kyoto Encyclopedia of Genes and Genomes (KEGG) is a database resource for understanding high-level functions and utilities of a biological system, such as the cell, the organism, and the ecosystem using molecular-level information, especially large-scale molecular datasets generated by genome sequencing and other high-throughput experimental technologies (http://www.genome.jp/kegg/, accessed on 18 June 2021). The ClusterProfiler R package tested the statistical enrichment of differentially expressed genes in the KEGG pathways [41]. Protein–protein interaction (PPI) analysis of DEGs was based on the STRING database, which is known and predicted by protein–protein interactions [42]. Further, PPI networks were visually analyzed using Cytoscape (version 3.8.2, San Diego, CA, USA) from Langmead et al. [43].

### 2.11. QRT-PCR Validation

The total RNA that had been extracted from the collected samples using cells was used for RNA extraction with RNAiso Plus (Takara, Dalian, China), as described in the manufacturer’s protocol. Complementary DNA (cDNA) synthesis was achieved using a HiFi Script gDNA Removal cDNA Synthesis Kit (CWBIO, Beijing, China). The quality and quantity of cDNA were adjusted using a NanoDrop 2000 (Thermo Fisher Scientific, Wilmington, DE, USA), and qRT-PCR was performed using a Light Cycler^®^ 96 (Roche, Basel, Switzerland) along with the primers listed in Appendix A.

The qRT-PCR reaction mixtures were prepared in triplicate using 10 μL of 2 × Power SYBR Green PCR Master Mixture (CWBIO, Beijing, China) containing 1 μL of each primer (10 μM), 1 μL of cDNA, and 7 μL of ddH_2_O. PCR amplification was performed under the following conditions: 95 °C for 20 s, followed by 45 cycles of 95 °C for 3 s and 61 °C for 30 s for dissociation curve analysis, then 95 °C for 15 s, 60 °C for 60 s, 95 °C for 15 s, and 60 °C for 15 s for cooling, and finally 37 °C for 30 s. The relative expression ratio of the target genes versus the β-actin (ACT) gene was calculated using the 2^−ΔΔCT^ method, and all data were presented in terms of the relative mRNA expression [44]. The experiment was repeated three times.

### 2.12. Statistical Analysis

All DEG data validated by qRT-PCR, with significant differences among means, were assessed by Tukey’s multiple range test, followed by one-way ANOVA (* *p* < 0.05, ** *p* < 0.01, *** *p* < 0.001). The data represented in all graphs are the means ± SDs from at least three biological replicates. Differences were considered to be significant at *p* < 0.05. All data were analyzed by GraphPad Prism software version 7.0 (GRAPH PAD Software Inc., San Diego, CA, USA). 

## 3. Results

### 3.1. SMs from B. methylotrophicus NJ13 Inhibited CBLJ-3

In vitro antifungal assays of strain CBLJ-3 were performed for 4 d and 7 d in NJ13 sterile filtrates with concentrations of 0.5%, 1%, 2.5%, 5%, and 10% (*v*/*v*). The *B. methylotrophicus* NJ13 sterile filtrates showed obvious antifungal activity against strain CBLJ-3, and as the concentration increased, the mycelial growth of strain CBLJ-3 was inhibited differently (Figure 1). Microscopic examination revealed that the form of strain CBLJ-3’s hyphae was round (Figure 2B) and branched (Figure 2F), and the leakage of the cell contents led to the formation of vacuolation (Figure 2E). Further, the cells of many hyphae had incomplete cell membranes and cell walls. However, these qualities were barely visible in the control group (Figure 2C).

### 3.2. Effect of NJ13’s SMs on the Germination of CBLJ-3’s Conidia

Through the cultivation of CBLJ-3 conidia on water agar containing different concentrations of NJ13 sterile filtrates (0.5%, 1%, 2.5%, 5%, and 10% (*v*/*v*)), the conidia germination rate of CBLJ-3 was 32.38–6.66%. As the concentration of NJ13 sterile filtrates increased, the spore germination rate of strain CBLJ-3 decreased significantly (Figure 3C). It was observed that strain CBLJ-3’s conidia in the treatment group were basically malformed, showing “dumb-bell-shaped” cells and vacuolation inside the cells, and observed by an optical microscope. 

### 3.3. Effect of NJ13’s SMs on the Cell Membrane of CBLJ-3

The extracellular conductivity of strain CBLJ-3 increased significantly when it was determined at six time points at concentrations from 10% to 70% of NJ13 sterile filtrates. Further, the extracellular conductivity of CBLJ-3 increased with the increase in the concentration of NJ13 sterile filtrates and the prolongation of the action time. After being treated with the 40% (*v*/*v*) concentration of NJ13 sterile filtrates for 24 h, CBLJ-3 recorded an extracellular conductivity value of 5.10 μs·cm^−1^. The highest extracellular conductivity value of 9.19 μs·cm^−1^ was detected in response to the 70% (*v*/*v*) concentration after 60 h (Figure 3D). Accordingly, the leakage of the cytoplasmic contents of strain CBLJ-3 was positively correlated with the concentrations of strain NJ13’s sterile filtrates.

As shown in Figure 4A,B, the MDA content of strain CBLJ-3 increased significantly (*p* < 0.001) after treatment with different sterile filtrate concentrations (EC_50_, 5%, 10%, and 20% (*v*/*v*)) of *B. methylotrophicus* NJ13. As the concentration of NJ1’s SMs increased, the MDA content of strain CBLJ-3 increased. When the mycelia of strain CBLJ-3 were treated with NJ13 sterile filtrates with a concentration of 20%, the highest content of MDA obtained was 3.98 nmol·g^−1^. Lipid peroxidation of strain CBLJ-3’s mycelia may be related to changes in the MDA content.

### 3.4. Glucose Absorption of Strain CBLJ-3 Was Affected by NJ13 Sterile Filtrates 

As shown in Figure 4C, the glucose content in the liquid medium decreased over time in the treatment groups. This suggested that strain CBLJ-3 continuously absorbed and utilized glucose during growth. The FS group’s glucose content of 11.35 mg·mL^−1^ was significantly higher than that of the FW group, 6.52 mg·mL^−1^, at 60 hpi in the PDB medium, indicating that NJ13 sterile filtrates inhibited the absorption of glucose by strain CBLJ-3.

### 3.5. In Vivo Test of Antifungal Activity in Panax ginseng Roots

In the bioassay of ginseng roots, no obvious lesions occurred in any root during the first 3 days of treatment. However, the roots started to turn yellow on the fifth day of infection with the pathogen, whereas this disease symptom was not observed in the roots sprayed with the NJ13 sterile filtrates. The disease symptoms of ginseng roots treated with *I. robusta* CBLJ-3 were noted from day 3 to day 9 at 2-day intervals. The yellowing and blackening of the fungal-treated roots were apparent, and these symptoms gradually worsened. In addition, the fungal mass on roots infected with *I. robusta* CBLJ-3 alone increased gradually, while no fungal growth was evident on roots treated with NJ13 sterile filtrates. These findings demonstrated moderate to excellent protective and curative effects of NJ13 sterile filtrates in vivo. At last, there was also a difference between ginseng roots with different times of the NJ13 sterile filtrates application, and ginseng roots when the NJ13 sterile filtrates were applied prior inoculation looked clearly better than those for which NJ13 sterile filtrates were applied after inoculation with the *I. robusta* CBLJ-3. The results are clearly shown in Figure 5.

### 3.6. Quality Control and Quantification of Gene Expression Levels 

The RNA-Seq analysis of strain CBLJ-3 was performed using NJ13 sterile filtrates. The sequencing results showed that the 12 filtered samples had more than 97% high-quality clean reads compared with the original reads. There were 6.93–7.43 G clean bases for each sample. These results are beneficial for data mining in bioinformatics. The bases with Phred values greater than 20 or 30 accounted for more than 97.8% and 93.73%, respectively, of the total bases (Q_20_ > 97.8% and Q_30_ > 93.73%), and the GC content was 55.02–55.84%. The mapping rate was higher than 97.78% for each sample (Appendix A).

### 3.7. Overall Analysis of Differentially Expressed Genes (DEGs)

The heat map of the gene expression patterns showed that there were most active in BF_R4d treated with *B. methylotrophicus* NJ13’s SMs for 4 dpi (Figure 6A), which indicated that more DEGs in BF_R4d responded to *B. methylotrophicus* NJ13’s SMs. Additionally, DEGs from different libraries, including BF_R4d vs. R4d and BF_R7d vs. R7d, were identified based on *p* adj < 0.05 (corrected *p*-value) and |log_2_(fold change)| ≥ 1. A total of 1960 DEGs (963 upregulated and 997 downregulated) were identified in the BF_R4d vs. R4d comparison (Figure 6B). However, only 354 DEGs (204 up-regulated and 150 downregulated) were present in the BF_R7d vs. R7d comparison (Figure 6C). As shown in Figure 6D, the overlapping part of the circles for BF_R4d vs. R4d and BF_R7d vs. R7d comprised 121 DEGs that might represent DEGs relevant to the induction of *B. methylotrophicus* NJ13’s SMs. Based on the Venn diagram analysis, 113 common DEGs (78 upregulated and 35 downregulated) were found between BF_R4d vs. R4d and BF_R7d vs. R7d (Figure 6E). Further, the expression levels of eight common DEGs were unique among the different samples (Figure 6F).

### 3.8. GO Enrichment Analysis of DEGs

Gene ontology (GO) can systematically annotate species genes and mainly includes three branches: cellular component (CC), molecular function (MF), and biological process (BP). In the comparison of BF_R4d vs. R4d (Figure 7A), the 1960 DEGs were significantly enriched in 11 GO terms (*p* adj ≤ 0.05) and were mostly involved in the molecular function category. In this category, the 189 DEGs were involved in nine terms: carbon-oxygen lyase activity (11 DEGs, 9 upregulated and 2 downregulated), carbohydrate binding (10 DEGs, 9 upregulated and 1 downregulated), coenzyme binding (63 DEGs, 35 upregulated and 28 downregulated), hydrolase activity acting on glycosyl bonds (38 DEGs, 29 upregulated and 9 downregulated), polysaccharide binding (7 upregulated DEGs), pattern binding (7 upregulated DEGs), hydrolase activity hydrolyzing O-glycosyl compounds (38 DEGs, 29 upregulated and 9 downregulated), cellulose binding (7 upregulated DEGs) and carbon-oxygen lyase activity acting on polysaccharides (8 upregulated DEGs). In the cellular component category, 14 DEGs were enriched in the extracellular region. Moreover, 52 DEGs (36 upregulated and 16 downregulated) were enriched in the carbohydrate metabolic processes, which belonged to the biological process category. However, when *p* adj < 0.05, DEGs were not enriched in any terms in BF_R7d vs. R7d (Figure 7B).

The 78 upregulated DEGs in overlapping circles (BF_R4d vs. R4d_up and BF_R7d vs. R7d_down) were enriched in 11 terms, i.e., 10 molecular function terms and only 1 cellular component term (Figure 7C). These terms were endopeptidase activity (3 DEGs); oxidoreductase activity acting on the CH-NH2 group of donors (2 DEGs); oxidoreductase activity acting on the CH-NH_2_ group of donors; oxygen as acceptor (2 DEGs); quinone binding (2 DEGs); primary amine oxidase activity (2 DEGs); polysaccharide binding (2 DEGs); pattern binding (2 DEGs); cellulose binding (2 DEGs); carbon-oxygen lyase activity acting on polysaccharides (3 DEGs); carbohydrate binding; and extracellular region (3 DEGs). In contrast, there were 35 downregulated DEGs in overlapping circles (BF_R4d vs. R4d_down and BF_R7d vs. R7d_down) that were significantly enriched, and three terms belonged to the molecular function category. In Figure 7C, the six downregulated DEGs were assigned to the following terms: antioxidant activity (2 DEGs), oxidoreductase activity acting on peroxide as acceptor (2 DEGs), and peroxidase activity (2 DEGs).

### 3.9. KEGG Enrichment Analysis of DEGs

To further explain gene function, all DEGs were enriched in different KEGG metabolic pathways, which can be used to analyze the molecular interaction network and record the unique metabolic form of an organism. Based on the k-means clustering algorithm, the 2193 DEGs were hierarchically clustered into four subclusters (Appendix A). Significant enrichment of the KEGG metabolic pathway was identified by DEG co-expression cluster analysis. Then, KEGG enrichment analysis was performed on the DEGs of each subcluster with *p* adj < 0.05. According to subclusters 2 and 3, *B. methylotrophicus* NJ13’s SMs induced the pathways of vitamin B6 metabolism, steroid biosynthesis, pentose and glucuronate interconversions, fatty acid degradation, tyrosine metabolism, and glycolysis/gluconeogenesis in strain CBLJ-3 at 4 dpi. Furthermore, *B. methylotrophicus* NJ13’s SMs significantly inhibited glycerolipid metabolism, glycerophospholipid metabolism, and lysine degradation in subcluster 1 of strain CBLJ-3 at 4 dpi. 

As shown in Figure 8, in the comparison of BF_R4d vs. R4d, 1960 DEGs (BF_R4d vs. R4d) were mapped to relative metabolism pathways that consisted of tryptophan metabolism (14 DEGs, 5 upregulated and 9 downregulated), taurine and hypotaurine metabolism (4 DEGs, 1 upregulated and 3 downregulated), vitamin B6 metabolism (5 upregulated DEGs), fatty acid degradation (10 DEGs, 6 upregulated and 4 downregulated), alanine, aspartate, glutamate metabolism (11 DEGs, 4 upregulated and 7 downregulated), tyrosine metabolism (18 DEGs, 10 upregulated and 8 downregulated), β-alanine metabolism (12 DEGs, 6 upregulated and 6 downregulated), glycolysis/gluconeogenesis (15 DEGs, 9 upregulated and 6 downregulated), steroid biosynthesis (10 upregulated DEGs), glycerolipid metabolism (13 DEGs, 5 upregulated and 8 downregulated), ABC transporters (13 DEGs, 7 upregulated and 6 downregulated), and pentose and glucuronate interconversions (21 DEGs, 16 upregulated and 5 downregulated). There were no terms with *p* adj < 0.05 in the comparison of BF_R7d vs. R7d. Only one term, tyrosine metabolism, was significantly enriched among the two groups (BF_R4d vs. R4d_up and BF_R7d vs. R7d_up) of common upregulated DEGs.

### 3.10. Analysis of DEGs in Response to Stress Caused by B. methylotrophicus NJ13’s SMs 

Next, we sought to understand the response mechanism of strain CBLJ-3 to biological signals and energy metabolism under stress caused by NJ13 sterile filtrates. Based on information from the STRING database, different groups (BF_R4d vs. R4d and BF_R7d vs. R7d) of DEGs were placed in the PPI network and analyzed using CytoNCA, a plugin for Cytoscape. Numerous nodes were aggregated together to form important modules. In the treatment with NJ13 sterile filtrates, four DEGs were hub genes in response to stress in CBLJ-3 (Appendix A and Appendix A). In this study, in screening the genes of CBLJ-3 that responded to *B. methylotrophicus* NJ13’s SMs, some DEGs were classified by analyzing the gene descriptions of the transcriptomic annotations. The DEGs were mainly divided into the following five categories: the glycosyl hydrolase family, sugar (and other) transporters, the cytochrome P450 enzyme superfamily, ABC transporters, and the major facilitator superfamily (Appendix A). The DEGs of the glycosyl hydrolase family were active at 4 dpi, especially many of the upregulated DEGs, and the metabolic processes of sugar (and other) transporters were inhibited. By combining the transcriptomic annotations with the above analysis, the potential response genes of strain CBLJ-3 to *B. methylotrophicus* NJ13’s SMs were screened (Appendix A). We investigated DEGs that encoded glucose transporters in response to stress caused by *B. methylotrophicus* NJ13’s SMs. *B. methylotrophicus* NJ13’s SMs may have induced the gene expression of related DEGs to damage fungal cell walls. They may also have played a vital role in bringing about sterol synthesis/transfer and metabolite synthesis in the metabolic regulation of CBLJ-3. Of course, the use of *B. methylotrophicus* NJ13’s SMs may have contributed to inducing drug resistance in CBLJ-3. Finally, gene annotation and the fold change in hub genes after PPI screening analysis were investigated.

### 3.11. Validation of Transcriptomic Results Using QRT-PCR

To validate the transcriptomic sequencing results, six genes from different functional categories were selected, namely encoding endo-1,4-β-xylanase 1, highly reducing polyketide synthase *azaB*, acetohydroxy-acid synthase catalytic subunit *AHAS*, putative sterigmatocystin biosynthesis peroxidase *stcC*, multidrug-resistance protein *CDR2*, and pectate lyase (Appendix A). According to transcriptomic analysis, all were jointly regulated by stress caused by NJ13 sterile filtrates at 4 dpi and 7 dpi. Then, 15 genes were selected for temporal expression profiles. RNA isolated from mycelia treated with NJ13 sterile filtrates was analyzed by qRT-PCR at five different time points. The expression levels of DEGs changed at the different time points and significantly increased or decreased at 4 dpi and 5 dpi (Figure 9). The transcriptomic data of the genes were validated by qRT-PCR, and their expression trends correlated significantly with the transcriptomic results of CBLJ-3 stressed by NJ13 sterile filtrates. To evaluate the effect of treatment with different concentrations of NJ13 sterile filtrates on DEGs, eight DEGs were selected with the same expression trends at 4 dpi and 7 dpi (Figure 10).

## 4. Discussion

### 4.1. Effect of the Antifungal Activity of B. methylotrophicus NJ13’s SMs on CBLJ-3

It is well-known that the hyphal growth of filamentous fungi is necessary for pathogenicity [45]. After treatment with NJ13’s SMs, the growth of strain CBLJ-3 was effectively inhibited in vitro and in vivo (Figure 1 and Figure 5). The inhibitory effects of SMs against strain CBLJ-3’s hyphal growth and conidial germination were visible and increased significantly as the concentration increased (Figure 3B,C). The above results are similar to previous studies on biocontrol *Bacillus* spp. [46]. For example, *B. methylotrophicus* strain NKG-1 significantly inhibited the mycelial growth of phytopathogenic fungi [47]. Moreover, the sterile filtrates of *B. subtilis* strain TE3 inhibited the mycelial growth of *Bipolaris sorokiniana,* which causes spot blotch in wheat [48]. Further, *B. velezensis* XT1 showed strong inhibitory effects on the hyphal growth and conidial germination of *Alternaria alternata*, *F. oxysporum*, *Monilinia fructicola*, *Magnaporthe oryzae*, *Thanatephorus cucumeris,* and *Sclerotinia sclerotiorum*. Therefore, it is important to understand the interaction mode between *B. methylotrophicus* NJ13 and *I. robusta* CBLJ-3 for the development of *B. methylotrophicus* NJ13 as a biocontrol agent. 

### 4.2. Effects of B. methylotrophicus NJ13’s SMs on the Cellular Structure of CBLJ-3 

An abnormal swelling of the hyphae in *S. sclerotiorum* was found against *B. subtilis* EDR4 in a previous study [49]. Similarly, in our study, *B. methylotrophicus* NJ13’s SMs led to the swelling (Figure 3A) and branching of strain CBLJ-3 hyphae (Figure 2B,F), and further observation by optical microscopy showed that the intracellular contents of the hyphae and conidia of strain CBLJ-3 increased and leaked out (Figure 2E), resulting in vacuolation (Figure 2D,H). To further explore the phenotypic mechanism, electrical conductivities of strain CBLJ-3 were examined and were found to increase after treatment with different concentrations of NJ13 sterile filtrates (Figure 3D). Therefore, it can be inferred that *B. methylotrophicus* NJ13’s SMs may affect the integrity of the cell wall and cell membrane of strain CBLJ-3. It might be a common phenomenon that the sterile filtrates of *Bacillus* spp. affect the normal growth morphology of phytopathogenic fungal hyphae to induce cellular damage and apoptosis [50]. For example, a study showed that *B. amyloliquefaciens* JCK-12 produced cyclic lipopeptides that damaged the cell membrane of *F. graminearum*, which causes Fusarium head blight [51]. Interestingly, pelgipeptins produced by *Paenibacillus elgii* JCK1400 could disrupt the cell membranes of *B. cinerea* and *R. solani* without destroying their cell walls [52].

The cell walls of most fungi are mainly composed of glucans (including β-(1-3)-glucan, β-(1-4)-glucan, and β-(1-6)-glucan), chitin, and glycoproteins, which are candidates for potential targeted drugs [53]. The components of most fungal cell walls are covalently cross-linked; for example, β-(1,3)-glucan is bound to chitin, and the interconnected β-(1,6)-glucan plays an important role in the structural organization of the cell wall [54]. Chitin synthesis-related genes (evm.TU.Contig19.330 and evm.TU.Contig1.1596) were down-regulated in the transcriptomic data in this study (Appendix A), and the synthesis of chitin, which constitutes a cell wall component, was inhibited under stress caused by NJ13 sterile filtrates. In recent years, glycoside hydrolases (GHs) have been studied in the process of plant–pathogenic fungal interactions [55]. Xyloglucanases (XEGs) can break down xyloglucan and have been studied primarily as virulence factors infecting the cell walls of plants [56]. Interestingly, this study found that many DEGs encoding GHs and three DEGs (evm.TU.Contig28.214, evm.TU.Contig19.313, and evm.TU.Contig28.103) encoding XEGs were upregulated after treatment with NJ13 sterile filtrates (Appendix A and Appendix A), and these filtrates may have induced other metabolic processes of CBLJ-3. For example, in *Trichoderma* species, a study showed that endoglucanases and chitinases play important roles in cell wall degradation of plant pathogens [57]. In the transcriptomic annotations (Appendix A), four DEGs encoding endo-β-1,4-glucanase were significantly upregulated at 4 dpi. Of course, under stress caused by NJ13 sterile filtrates, evm.TU.Contig18.225 (endo-β-1,4-glucanase B) and evm.TU.Contig4.662 (endo-1,4-beta-xylanase 1) were upregulated at different times and concentrations based on qRT-PCR (Figure 9 and Figure 10). Meanwhile, three upregulated DEGs that encoded chitinase-1 (glycosyl hydrolases family 18) were recorded in the transcriptomic results. In the previous study, *B. subtilis* strain ALICA could express two hydrolytic enzymes, β-1,4-glucanase and protease, which possibly degrade the contents of the cell walls of *Collegium gloeosporioides*, *A. alternata*, *Macrophomina* sp., *B. cinerea*, and *Sclerotium rolfesii*, such as β-1,3-glucan and glucosidic bonds [58].

The fungal cell membrane mainly consists of three components, namely sterols, glycerophospholipids, and sphingolipids [59]. The ergosterol biosynthesis gene is highly conserved and complexly regulated. Therefore, the regulation (including ergosterol gene overexpression and ergosterol gene deletion) of the ergosterol biosynthesis gene is critical for affecting cell wall synthesis [60]. In our study, four upregulated DEGs were screened and annotated with ergosterol-biosynthesis-related genes in the transcriptomic results (Appendix A), and evm.TU.Contig23.290 (sterol-C5-desaturase) was upregulated based on the qRT-PCR analysis (Figure 9 and Figure 10). Sterol uptake control protein 2 (*UPC2*) is an important transcriptional regulator of ergosterol synthesis and plays a regulatory role in response to environmental changes [61]. In pathogenic fungi, the deletion of *UPC2* may result in significantly reduced tolerance to azole drugs, and the deletion of *UPC2* is lethal in *Saccharomyces cerevisiae* [62]. Similarly, in our study, two upregulated DEGs that annotated *UPC2* were induced at 4 dpi and 7 dpi by treatment with NJ13 sterile filtrates (Appendix A). A previous study indicated that under the catalysis of UDP-glucose, sterol glucosyltransferase used ergosterol as a sugar receptor to form sterol glycosides, which are important parts of eukaryotic cell membrane glycolipids [63]. In our study, five downregulated DEGs encoded UDP-glucose, which were found to be sterol glucosyltransferases in the transcriptomic annotations (Appendix A). Based on qRT-PCR, they were analyzed for the validation of gene expression profiles (Figure 9 and Figure 10). It was found that after treatment with NJ13 sterile filtrates, the synthesis of sterol in strain CBLJ-3 was disordered, and stress-responsive genes also evolved to maintain their own metabolic function. Moreover, the dynamic changes in the MDA content suggested that NJ13 sterile filtrates not only inhibited the biosynthesis of sterols but also accelerated the lipid peroxidation of strain CBLJ-3 (Figure 4A,B). Thus, it was further confirmed that the membrane structure and function of strain CBLJ-3 were severely damaged.

### 4.3. Analysis of the Effect of B. methylotrophicus NJ13’s SMs on CBLJ-3’s Metabolism

Glucose transporters may promote the transport of glucose (and related substances) across the cell membrane, and the glucose transport system of microorganisms is a conservative strategy for responding to nutrient disturbances from the external environment [64]. *GRT2* and *SNF3* are involved in intracellular glucose transport as glucose-sensing receptors in *Saccharomyces cerevisiae* [65]. *HXT1-7* are the most important glucose transporters in *S. cerevisiae* [66]. In addition, surfactin extracted from *Brevibacillus brevis* KN8(2) was able to inhibit the DNA and protein metabolism of *F. moniliforme* [67]. In our study, transcriptomic analysis of strain CBLJ-3 in the presence of NJ13 sterile filtrates (Appendix A) revealed downregulated DEGs that encoded glucose transporters. Biochemical experiments also showed that glucose availability in strain CBLJ-3 was significantly reduced in response to treatment with NJ13 sterile filtrates (Figure 4C). It is speculated that these filtrates might affect the glucose metabolism of strain CBLJ-3 by inhibiting genes that encode glucose transporters.

Some SMs of fungi, such as fusarin produced by *Fusarium* [68], can participate in the process of infecting the host as a toxin, which increases the damage to the host. Therefore, genes related to toxin synthesis are particularly important in pathogenic fungi–plant interactions. For example, most *Fusarium* species can produce fusarin C and *FUS1* as a hub gene in fusarin biosynthesis [69]. At the same time, the pathogenicity of *Magnaporthe oryzae Triticum* (MoT) is closely related to apppressorium melanization, and lipopeptides produced from *B. subtilis* strain 109GGC020 could induce abnormal appressoria (low melanization), which inhibits the infection process of MoT [70]. It was also found in this study that some DEGs related to the synthesis of secondary metabolites of strain CBLJ-3 were downregulated, among which three DEGs are involved in the biosynthesis of T-toxin, fusarin C, and destruxin [71,72] (Appendix A). We speculated that NJ13 sterile filtrates may have inhibited the expression of genes related to strain CBLJ-3 toxin synthesis, and the secretion of strain CBLJ-3’s SMs may have been disturbed.

The overexpression of genes encoding ABC transporters and the major facilitator superfamily (MFS) are among the major mechanisms of fungal drug resistance [73,74], enabling fungi to develop resistance to many structurally and functionally diverse toxicants. ABC transporters, including Candida resistance 1 protein (Cdr1p), are important membrane transporters in *Candida albicans* that may maintain the resistance of pathogens to antifungal drugs [75]. Upon the stress of *B. amyloliquefaciens* strain SDF-005, some upregulated DEGs might enhance the ability of efflux pumps that transport exogenous antimicrobial substances out of the cells of *Colletotrichum gloeosporioides* sensu stricto (s.s.) TS-09R, reducing accumulation and minimizing damage caused to cells [76]. Interestingly, a previous study showed that *Zymoseptoria tritici*, which causes leaf spots in *Septoria*, exhibited strong resistance to fungicides, and the inactivation of the MFS gene (*MgMFS1*) abolished resistance to fungicides in *Z. tritici*. [77]. In upregulated resistance-related genes of strain CBLJ-3, which were treated with *B. methylotrophicus* NJ13’s SMs (Appendix A), four upregulated DEGs encoding ABC transporters of efflux pumps’ CDR were found, of which only *CDR1* and *CDR2* have been reported as critical drug-resistance genes [78]. Further, four DEGs encoding the major facilitator superfamily were upregulated at different time points based on qRT-PCR (Figure 9). In addition, evm.TU.Contig 1.616 (major facilitator superfamily), evm.TU.Contig 5.810 (multidrug-resistance protein CDR2), and evm.TU.Contig 17.77 (major facilitator superfamily) were upregulated at different concentrations of NJ13 sterile filtrates (Figure 10). However, the regulatory mechanisms of drug-resistance-related genes in strain CBLJ-3 still need to be further verified.

### 4.4. Unanswered Questions and Future Possibilities

As seen in Figure 11, in this study, the response strategies of strain CBLJ-3 to strain NJ13’s SMs were inferred. However, some unanswered questions remain.

Firstly, due to the complexity of the substances in NJ13 sterile filtrates, this study could not determine the different components of antifungal substances that might interact with intracellular targets in strain CBLJ-3. It is necessary to purify the antifungal product produced by strain NJ13 and further characterize its chemical structure. Therefore, future work will focus on finding potential targets in strain CBLJ-3 that interact with single components in *B. methylotrophicus* NJ13’s SMs to clarify the biocontrol mechanism of strain NJ13.

Secondly, ROS are present in cells as byproducts of metabolic processes in organisms, and organisms must have ROS homeostasis to maintain normal metabolism [79]. Fengycins isolated from cyclic lipopeptides of *Bacillus subtilis* have been found to induce ROS that induce apoptosis in *Magnaporthe grisea*, and the organelles of *M. grisea* have been found to be damaged based on transmission electron microscopy. Further, the response of phytopathogenic fungi to DNA damage has been found to be accompanied by DNA repair function, according to proteomic analysis [80]. Honokiol may also induce the accumulation of ROS to damage mitochondria in *B. cinerea*, and it might also induce autophagy and apoptosis [81]. However, our work may not be supportive of the theory that *B. methylotrophicus* NJ13’s SMs can induce the accumulation of ROS in strain CBLJ-3.

Finally, it has been reported that autophagy in fungi usually occurs in response to environmental stress [82]. In general, the autophagy pathway may degrade senescent and dead organelles and proteins in cells [83]. Regarding nutrient deprivation, vacuoles are degradative organelles, and autophagy in filamentous fungi is accompanied by the enlargement and degradation of vacuoles in the hyphae, with autophagosomes being found in the vacuoles [84]. Further, a study on *Cylindrocarpon destructans* showed that the active substance of *Trichoderma atroviride* T2 induced autophagy in pathogenic fungi [85]. In our study, under stress caused by *B. methylotrophicus* NJ13’s SMs, autophagy may have occurred in strain CBLJ-3. Currently, we are not able to demonstrate the direct relationship between the occurrence of autophagy and apoptosis in strain CBLJ-3. Under stress caused by *B. methylotrophicus* NJ13’s SMs, exceeding the intensity threshold of autophagy in strain CBLJ-3 is the key to inducing apoptosis [86]. The exploration of autophagy in strain CBLJ-3 might be the key to controlling ginseng rusty root rot in the future.

Ginseng rhizosphere endophytic bacteria, including *Bacillus* spp., showed strong antagonistic activity against phytopathogenic fungi [3,87]. The *Bacillus* spp. has been extensively studied for biological control of phytopathogenic fungi over the decades [88,89,90]. At the same time, with the deepening of research, the biological control effect of *B. methylotrophicus* has also been reported [91,92,93]. In the Figure 5, the effects of *B. methylotrophicus* NJ13 on ginseng rusty root rot were showed in an in vivo experiment, which provided a good potential for application of *B. methylotrophicus* NJ13. Therefore, *B. methylotrophicus* NJ13 could be become an alternative for chemical fungicides as a novel biological control agent in the future.

## Figures and Tables

**Figure 1 jof-08-00779-f001:**
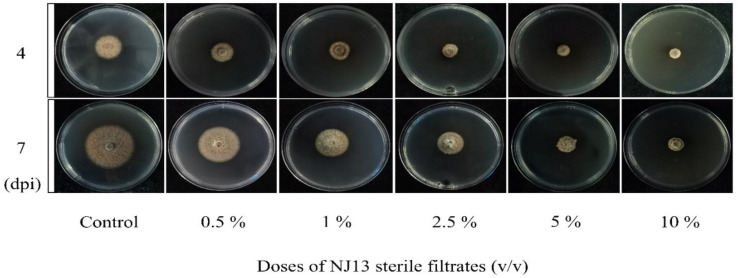
Antifungal activity of NJ13 sterile filtrates against CBLJ-3.

**Figure 2 jof-08-00779-f002:**
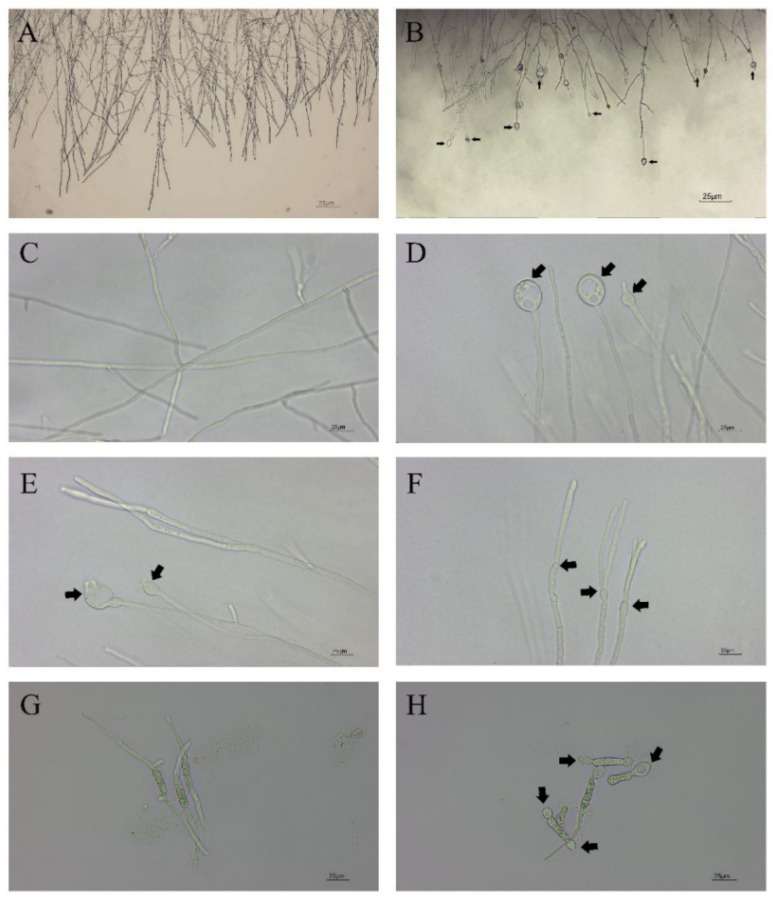
Hyphal and conidial morphology of CBLJ-3 photographed by optical microscope. These black arrows point to the deformed parts. (**A**,**C**) Straight mycelium, normal branching, normal septation; (**B**) hyphal swellings; (**D**) mycelial vacuolations; (**E**) content leakage; (**F**) mycelial branching; (**G**) normal germination of spores; (**H**) dumbbell-shaped megaspores. Bar = 25 μm.

**Figure 3 jof-08-00779-f003:**
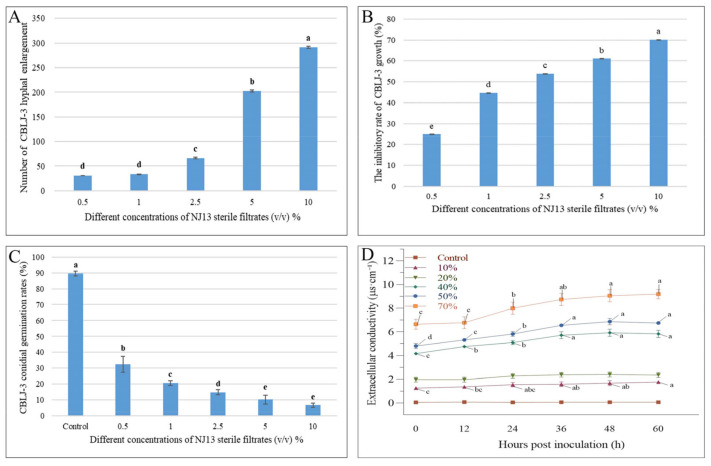
Effects of NJ13 sterile filtrates on CBLJ-3 (Means of significant values are separated by Fisher’s Least Significant Difference test (LSD) (*p* = 0.05), and denoted by lowercase letters.). (**A**) The number of enlarged mycelial cells. (**B**) Effect of NJ13 sterile filtrates on conidial germination of CBLJ-3. (**C**) Inhibition of mycelial growth of CBLJ-3 by NJ13 sterile filtrates. (**D**) Effect of NJ13 sterile filtrates treatments on the extracellular conductivity.

**Figure 4 jof-08-00779-f004:**
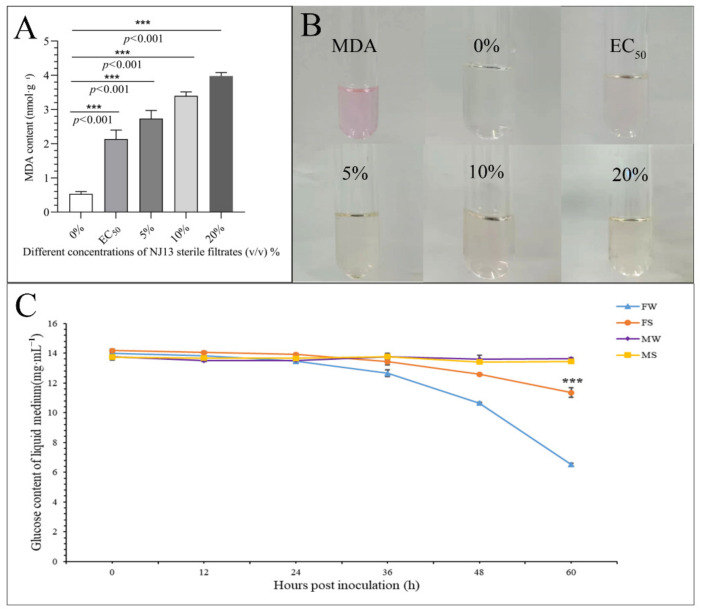
Determination of physiological indicators of CBLJ-3 affected by NJ13 sterile filtrates. (**A**) Determination of MDA content in CBLJ-3 mycelium. (**B**) Color features of MDA with different treatments. (**C**) Glucose content of CBLJ-3 liquid medium. FW, positive control; FS, treatment (compared with FW, *** *p* < 0.001); MW and MS, negative control.

**Figure 5 jof-08-00779-f005:**
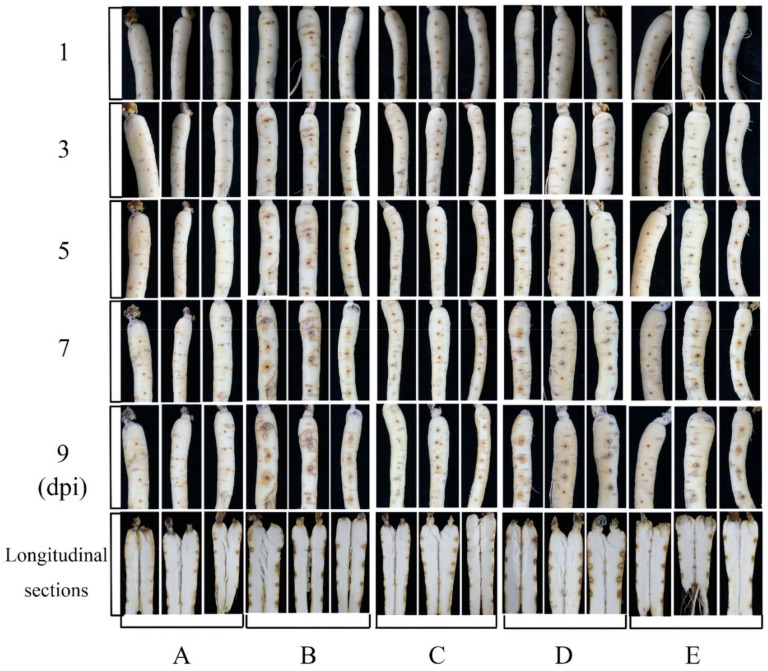
The antifungal activity of NJ13 sterile filtrates in vitro inoculation on detached *Panax ginseng* roots. (**A**) The blank control. (**B**) The strain CBLJ-3 was inoculated after one day of the application of sterile water in ginseng roots. (**C**) The strain CBLJ-3 was inoculated after one day of the application of NJ13 sterile filtrates in ginseng roots. (**D**) Applied sterile water after one day of inoculation with strain CBLJ-3 in ginseng roots. (**E**) Applied NJ13 sterile filtrates after one day of inoculation with strain CBLJ-3 in ginseng roots.

**Figure 6 jof-08-00779-f006:**
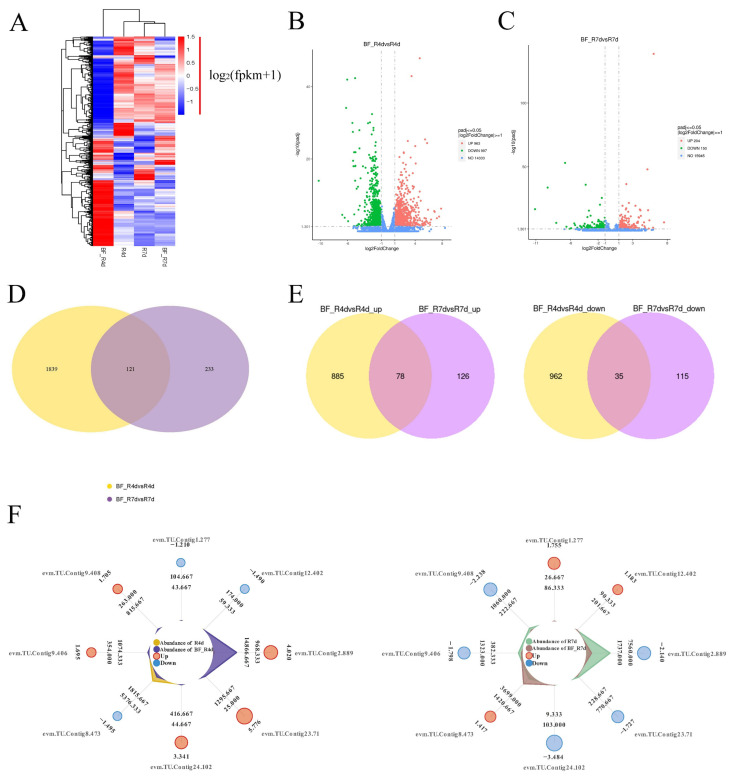
Analysis of DEGs in different samples. (**A**) Heat map analysis of samples from different treatments and time. Genes or samples with similar expression patterns in the heat map would be clustered together. The redder, the higher the gene expression level, and the bluer, the lower the gene expression level. White showed that the gene expression does not change. log_2_(fpkm + 1) from −1.5 to 1.5. Volcano plot of DEGs comparing (**B**) BF_R4d vs. R4d and (**C**) BF_R7d vs. R7d. The X-axis indicates log_2_(fold change) of DEGs among two samples. The Y-axis indicates the −log_10_(*p* adj) (Corrected *p*-value as *p* adj, which was used to assess the significance of changes in DEGs.) of DEGs variations. Red, green, and blue represent up-regulated and down-regulated genes and unchanged genes, respectively. Venn diagram of DEGs showed in groups, including (**D**) BF_R4d vs. R4d and BF_R7d vs. R7d, (**E**) BF_R4d vs. R4d_up and BF_R7d vs. R7d up, and BF_R4d vs. R4d_down and BF_R7d vs. R7d down. (**F**) Radar charts analysis of eight common DEGs in different samples. (Cycles 1 (from outside to inside) is gene identification. Cycles 2 is log_2_(fold change). Cycles 4 and 5 are the read counts for control and treatment, respectively).

**Figure 7 jof-08-00779-f007:**
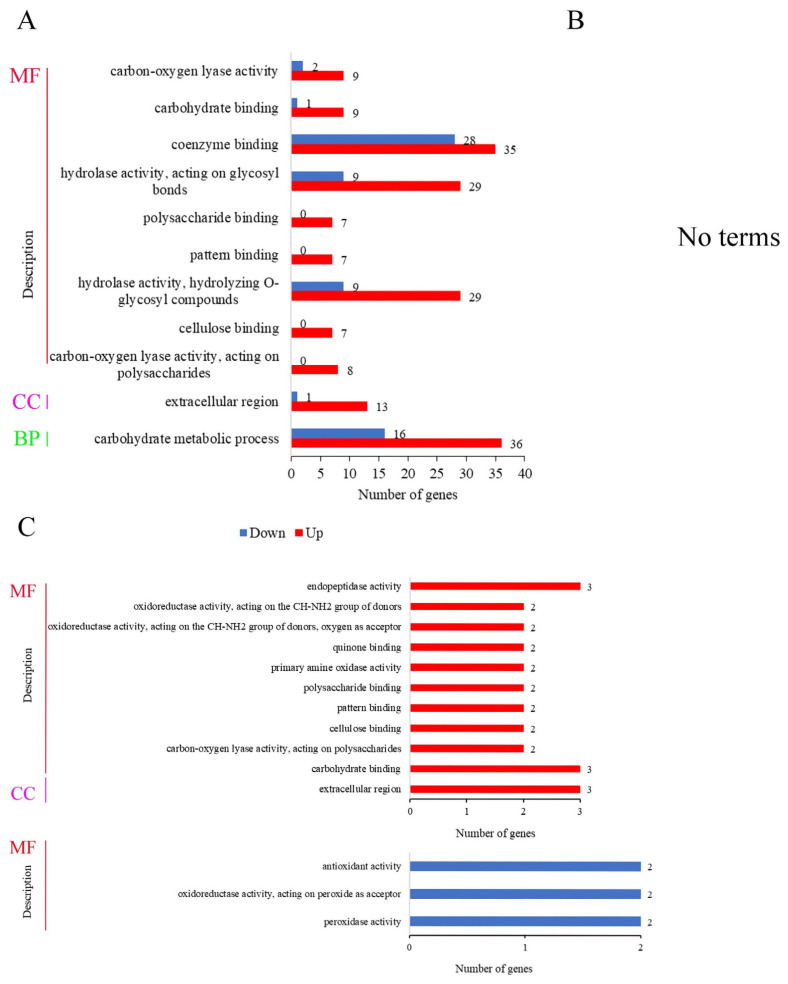
GO function analysis of DEGs at 4 dpi and 7 dpi (*p* adj < 0.05). MF, molecular function; CC, cellular component; BP, biological process. (**A**) Number of DEGs in GO function enrichment at 4 dpi. (**B**) No terms were enriched at 7 dpi. (**C**) GO function enrichment analysis of common upregulated DEGs and downregulated DEGs between BF_R4d vs. R4d and BF_R7d vs. R7d.

**Figure 8 jof-08-00779-f008:**
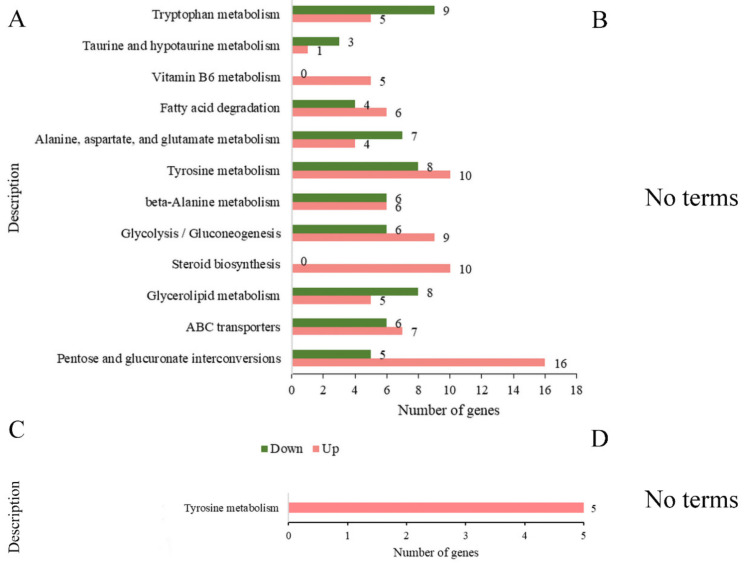
KEGG enrichment analysis of DEGs at 4 dpi and 7 dpi (*p* adj < 0.05). (**A**) Number of DEGs in KEGG enrichment at 4 dpi. (**B**) When *p* adj < 0.05, the samples of 7 dpi were not enriched in any terms. KEGG enrichment analysis of common (**C**) upregulated DEGs and (**D**) down-regulated DEGs between BF_R4d vs. R4d and BF_R7d vs. R7d. (**D**) When *p* adj < 0.05, common downregulated DEGs were not enriched in any terms.

**Figure 9 jof-08-00779-f009:**
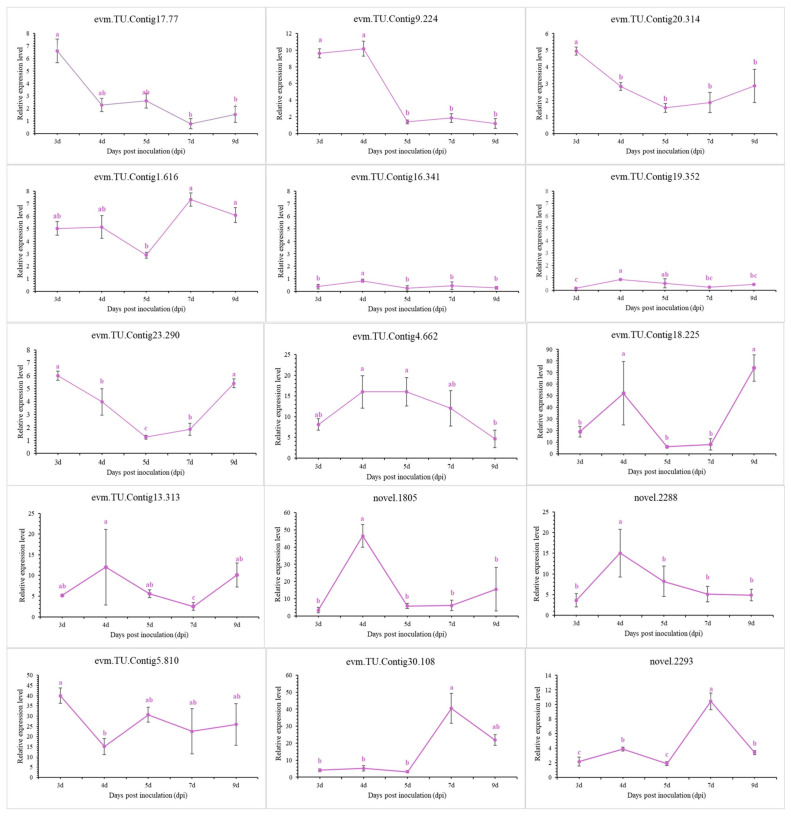
Analysis of gene expression profiles under NJ13 sterile filtrates stress. Tissue samples were taken at different intervals (3, 4, 5, 7, and 9 dpi) and were analyzed through qRT-PCR (Means of significant levels are separated by Fisher’s Least Significant Difference test (LSD) (*p* = 0.05), and denoted by lowercase letters.), including evm.TU.Contig17.77 (major facilitator superfamily), evm.TU.Contig9.224 (major facilitator superfamily), evm.TU.Contig20.314 (major facilitator superfamily), evm.TU.Contig1.616 (major facilitator superfamily), evm.TU.Contig16.341 (sterol 3-beta-glucosyltransferase UGT80A2), evm.TU.Contig19.352 (sterol-3-beta-glucosyltransferase UGT80A2), evm.TU.Contig23.290 (sterol-C5-desaturase), evm.TU.Contig4.662 (endo-1,4-beta-xylanase 1), evm.TU.Contig18.225 (endo-beta-1,4-glucanase B), evm.TU.Cotig13.313 (pectate lyase), novel.1805 (amino acid permease), novel.2288 (pectate lyase), evm.TU.Contig5.810 (multidrug-resistance protein *CDR2*), evm.TU.Contig30.108 (pectate lyase), and novel.2293 (pectate lyase).

**Figure 10 jof-08-00779-f010:**
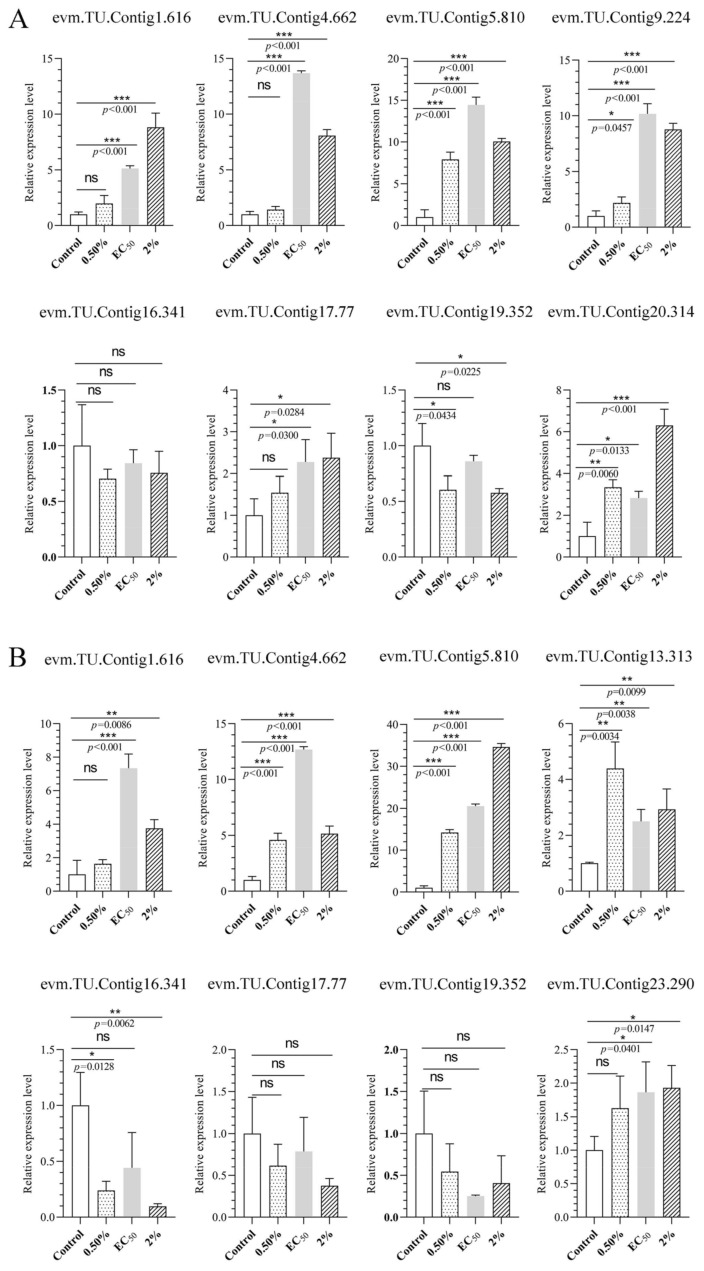
Analysis of gene expression profiles under different concentrations of NJ13 sterile filtrates stressed at (**A**) 4 dpi and (**B**) 7 dpi by qRT-PCR (* *p* < 0.05, ** *p* < 0.01, *** *p* < 0.001, and ns: not significant.), including evm.TU.Contig 1.616 (major facilitator superfamily), evm.TU.Contig 4.662 (endo-1,4-beta-xylanase 1), evm.TU.Contig 5.810 (multidrug-resistance protein CDR2), evm.TU.Contig 13.313 (pectate lyase), evm.TU.Contig 16.341 (sterol 3-beta-glucosyltransferase UGT80A2), evm.TU.Contig 17.77 (major facilitator superfamily), evm.TU.Contig 19.352 (sterol 3-beta-glucosyltransferase UGT80A2), and evm.TU.Contig 23.290 (sterol-C5-desaturase).

**Figure 11 jof-08-00779-f011:**
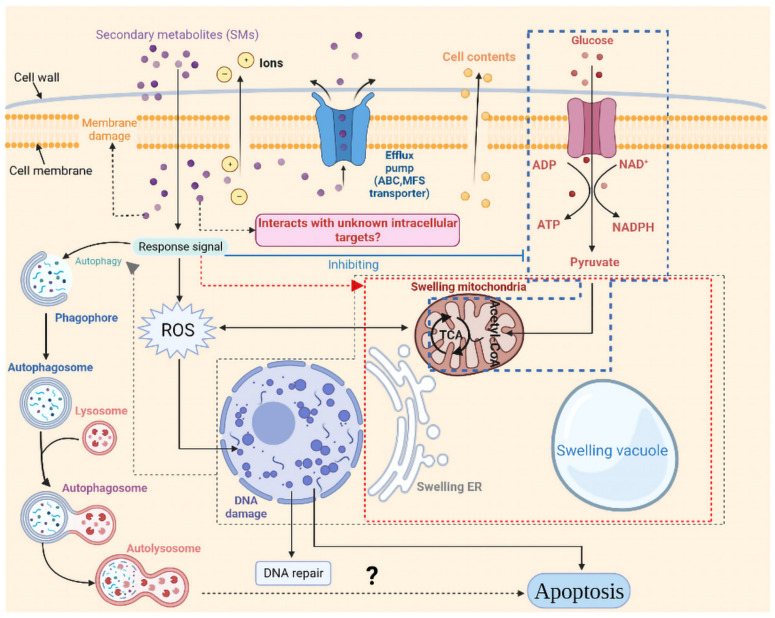
Response pattern of *I. robusta* CBLJ-3 to *Bacillus methylotrophicus* NJ13 sterile filtrates stress. Created with BioRender.com (https://biorender.com, accessed on 9 April 2020).

## Data Availability

Not applicable.

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
