# Peer review of "RNA-Seq Provides Insights into the Mechanisms Underlying Ilyonectria robusta Responding to Secondary Metabolites of Bacillus methylotrophicus NJ13"

_jof, 2022, doi:10.3390/jof8080779_

Round 1
Reviewer 1 Report
The authors have conducted a series of experiments to show the physiological effects of Bacillus strain NJ13 on the growth inhibition of the fungal ginseng root rot. Even more interesting, the authors also elucidated possible crucial pathways that the Bacillus used to inhibit the fungal pathogen. All of the methods, results and discussion are well-organized and impressive. However, there is a few points that can be clarified to this manuscript. A few minutes of revision should have this in top shape.
1.) Why was NJ13 selected to used in this study? Is there any preliminary result showing its potential to negatively affect pathogenicity of CBLJ-3?
2.) In the authors' point of view, effects of NJ13's SMs on CBLJ-3's cell structure and metabolisms discovered in this work can be implied to those of other Bacillus strains or not? Please discuss.
3.) Please suggest possible application of strain NJ13 as a biocontrol of ginseng root rot.
Author Response
Response to Reviewer 1 Comments
Point 1: Why was NJ13 selected to used in this study? Is there any preliminary result showing its potential to negatively affect pathogenicity of CBLJ-3?
Response 1: The previous studies published by our lab have shown that the NJ13 strain isolated from the stem of ginseng has broad-spectrum resistance to the pathogenic fungi of ginseng. The content of this section has been supplemented to the introduction.
(Line 83-91: “The results of previous studies that an endophytic B. methylotrophicus NJ13 isolated from the ginseng stem was chosen, which exhibited broad-spectrum antago-nism to pathogenic fungi including F. solani and Alternaria panax in ginseng, and showed a better control effect on Alternaria leaf spot of ginseng in the field [30-32]. At the same time, Secondary metabolite gene clusters of strain NJ13 were predicted by genomic sequencing and the results showing its potential to negatively affect pathogenicity of I. robusta CBLJ-3 [33]. And an antifungal protein, FlgG, that had antagonistic activity against I. robusta was identified, and it was proven that the production of secondary metabolic antibiotics was one of the main biocontrol mechanisms for strain NJ13 [34].”)
Point 2: In the authors' point of view, effects of NJ13's SMs on CBLJ-3's cell structure and metabolisms discovered in this work can be implied to those of other Bacillus strains or not? Please discuss.
Response 2: Thanks to the reviewer for valuable comments on the discussion of the article, which will be helpful for the improvement of the discussion. The content of this section has been supplemented to the discussion.
(Line 471-475: ”For example, a study showed that B. amyloliquefaciens JCK-12 produced cyclic lipopep-tides that damaged the cell membrane of F. graminearum, which causes Fusarium head blight [50]. Interestingly, pelgipeptins produced by Paenibacillus elgii JCK1400 could disrupt the cell membranes of B. cinerea and R. solani without destroying their cell walls [51].”
Line 501-505: “In the previous study, B. subtilis strain ALICA could express two hydrolytic enzymes, β-1,4-glucanase and protease, which possibly degrade the contents of the cell walls of Collegium gloeosporioides, A. alternata, Macrophomina sp., B. cinerea, and Sclerotium rolfesii, such as β-1,3-glucan and glucosidic bonds [57].”
Line 537-539: ”In addition, surfactin extracted from Brevibacillus brevis KN8(2) was able to inhibit the DNA and protein metabolism of F. moniliforme [66].”
Line 549-552: ”At the same time, the pathogenicity of Magnaporthe oryzae Triticum (MoT) is closely related to apppressorium melanization, and lipopeptides produced from B. subtilis strain 109GGC020 could induce abnormal appressoria (low melanization), which inhibited the infection process of MoT [69].”
Line 563-566: “Upon the stress of B. amyloliquefaciens strain SDF-005, some upregulated DEGs might enhance the ability of efflux pumps that transport exogenous antimicrobial substances out of the cells of Colletotrichum gloeosporioides s.s. TS-09R, reducing accumulation and minimizing damage caused to cells [75].”)
Point 3: Please suggest possible application of strain NJ13 as a biocontrol of ginseng root rot.
Response 3: Thanks to the reviewer for a valuable comment. We have supplemented that strain NJ13 may be used for biocontrol of ginseng rusty root rot.
(Line 616-624: “Ginseng rhizosphere endophytic bacteria including Bacillus spp. showed strong antagonistic activity against phytopathogenic fungi [86,87]. The Bacillus spp. has been extensively studied for biological control of phytopathogenic fungi over the decades [88-90]. At the same time, with the deepening of research, the biological control effect of B. methylotrophicus has also been reported [91-93]. In the Figure 5, the effects of B. methylotrophicus NJ13 on ginseng rusty root rot were showen in in vivo experiment, which provided a good potential for application of B. methylotrophicus NJ13. Therefore, B. methylotrophicus NJ13 could be become a altertive for chemical fungicides as a novel biological control agent in the future.”

Reviewer 2 Report
Authors presented a good and well-organized study with a large volume of data to elucidate the antifungal effect of the studied NJ13 strain on one of fungal pathogens of ginseng. The study combines cytological, physiological, and genomic evaluation of the antifungal action of NJ13. The obtained results and their interpretation can be useful for other researchers working in this field and may contribute into the understanding of fundamental bases of the revealed effect as well as be useful for the future development of the corresponding antifungal preparations. Of course, the further determination of the active component of the culture filtrate is required.
I have some minor comments concerning the use of some terms and description of methodology. After their correction, the paper can be published.
Introduction
Line 39-41: please check the grammar: you start from the singular “It”, but continue with plural “are considered”. Please, check also across the whole manuscript.
Line 49-50: fungal SMs also have a wide range of application in the biocontrol of plant pathogens. The same is true for actinomycetes.
Materials and methods
Line 98: I suggest it is necessary to insert “and incubated” after the word “medium, since now it looks like if you performed inoculation for 7 days.
Line 100-101: please, check the sentence. Do you mean the strain was first incubated in the LB broth overnight, then was transferred and incubated in a new fermentation medium? In the section 2.1 you mention the strain was grown on LB agar for two days; please, describe in details how and at which amount it was transferred to the liquid broth.
Line 106: what was the storage time (until use) for the filtrate?
Line 107: “dual culture assay” means the co-culturing of a pathogen and antagonistic strain. In your case (cultivation of a pathogen on agar medium supplemented with the cell-free NJ13 filtrate, it should be rather “antifungal activity assay”.
Subsection 2.5: just to check the volumes: you added 100 ul of spore suspension to 50 ml of PDB medium, then after a 2-day incubation centrifuged it (using which centrifuge?), washed and resuspended the sediment in other 50 ml of water supplemented with different filtrate concentrations. Is it correct?
Line 147: please indicate the kit.
Subsection 2.8: please, give more details in the experiment description to make it clear for anyone who would want to reproduce it. Was this experiment arranged on detached roots? How long time has passed between the detachment and inoculation? How did you inoculate mycelium plugs and SM? I suppose you used a pipette to drop SM at the wound (at which concentration(s)?), but how did you put mycelium plugs and how did you standardized them between the variants? Did you remove mycelial plugs from wounds prior SM application?
Line 210: please, explain the PPI abbreviation.
Results
Line 239: "antibiological" is inappropriate word. Probably you mean antifungal?
Line 241: antibacterial activity? Probably you mean "antifungal" or "antimicrobial"?
Figure 1: "antagonistic" is a wrong term as you do not use co-cultivation of two strains, but use culture filtrate of one of tem. I consider "antifungal" would be better.
Line 249: the "co-culturing" term is used in the case of use of two live cultures. Since you use culture filtrate of NJ13, this term can not be used. Please, replace it with a simple "cultivation". Also, the agar medium used does not consist of only culture filtrate, so I suggest it would be beter to replace "consisting of" with "containing" or "supplemented with".
Line 257: probably you mean "was measured" or "was determined" instead of "was treated"? Please, check.
Subsection 3.5: as I can see from the Fig. 5, there is also difference between the variants with different time of SM application; the variant when the SM was applied prior inoculation looks clearly better than that when it was applied after inoculation with the pathogen. I suggest it can be reflected in the text of this section.
Discussion
Fig. 11: it is incorrect to use the "antibacterial molecules" term in relation to fungal cells. Please, check and correct.
Line 541: I suggest "clarify" would be better than "illuminate".
Author Response
Response to Reviewer 2 Comments
Introduction
Point 1: Line 39-41: please check the grammar: you start from the singular “It”, but continue with plural “are considered”. Please, check also across the whole manuscript.
Response 1: Thanks to the reviewer's suggestion, and we have corrected the “is considered”, and also checked across the whole manuscript.
Point 2: Line 49-50: fungal SMs also have a wide range of application in the biocontrol of plant pathogens. The same is true for actinomycetes.
Response 2: Thanks to the reviewer for a valuable comment. And the section was rewritten for line 49-51 (“Therefore, it is necessary to explore biological control agents (BCAs) and secondary metabolites (SMs) produced by fungi, bacteria mainly including Bacillus spp., and actinomycetes as effective eco-friendly alternatives to chemical fungicides [12].”).
Materials and methods
Point 3: Line 98: I suggest it is necessary to insert “and incubated” after the word “medium, since now it looks like if you performed inoculation for 7 days.
Response 3: Thanks to the reviewer's suggestion, and we have inserted the “and incubated” in the line 108.
Point 4: Line 100-101: please, check the sentence. Do you mean the strain was first incubated in the LB broth overnight, then was transferred and incubated in a new fermentation medium? In the section 2.1 you mention the strain was grown on LB agar for two days; please, describe in details how and at which amount it was transferred to the liquid broth.
Response 4: Thanks to the reviewer for a valuable comment. And the section was rewritten for line 110-113 (“A agar disc of the strain NJ13 with a diameter of 8 mm was firstly inoculated into the LB broth overnight, and then was transferred, at inoculum volume of 10% (v/v), to 250 mL shake flasks with a working volume of 100 mL, and incubated in a new fermentation medium…”).
Point 5: Line 106: what was the storage time (until use) for the filtrate?
Response 5: Stored at 4 °C until use the next day. we have corrected the line 118 (“…and were stored at 4 °C until for use in the next day.”).
Point 6: Line 107: “dual culture assay” means the co-culturing of a pathogen and antagonistic strain. In your case (cultivation of a pathogen on agar medium supplemented with the cell-free NJ13 filtrate, it should be rather “antifungal activity assay”.
Response 6: Thanks to the reviewer's suggestion, and we have corrected the line 119 (“Antifungal Activity Assay…”).
Point 7: Subsection 2.5: just to check the volumes: you added 100 ul of spore suspension to 50 ml of PDB medium, then after a 2-day incubation centrifuged it (using which centrifuge?), washed and resuspended the sediment in other 50 ml of water supplemented with different filtrate concentrations. Is it correct?
Response 7: Yes, it is correct. Of course, we have checked the volumes. The content of this section has been supplemented to the subsection 2.5.
(Line 141-145: “Added 100 μL of above the spore suspension to 50 mL of PDB medium, then at 28 °C after a 2-day incubation. After being centrifuged (Eppendorf Centrifuge 5810R) at 8,000 rpm for 2 min, washed and resuspended the sediment in other 50 ml of water supplemented with different concentrations of NJ13 sterile filtrates (0%, 10%,20%, 40%, 50%, and 70% (v/v)). The mixture was centrifuged (Eppendorf Centrifuge 5810R)…”).
Point 8: Line 147: please indicate the kit.
Response 8: Thanks to the reviewer's suggestion, and we have indicated the line 160 (“…a MDA assay kit…”).
Point 9: Subsection 2.8: please, give more details in the experiment description to make it clear for anyone who would want to reproduce it. Was this experiment arranged on detached roots? How long time has passed between the detachment and inoculation? How did you inoculate mycelium plugs and SM? I suppose you used a pipette to drop SM at the wound (at which concentration(s)?), but how did you put mycelium plugs and how did you standardized them between the variants? Did you remove mycelial plugs from wounds prior SM application?
Response 9: Thanks to the reviewer's suggestion. We have given as much as possible more details in the experiment description to make it clear for anyone who would want to reproduce it. The content of this section has been supplemented to the subsection 2.8.
(Line 173-180: “Three-year-old detached ginseng roots were chosen for the in vivo experiment. Then, healthy roots were detached one day prior to inoculation, and were washed, surface disinfested with 75% alcohol, and rinsed with sterile water. Wounds with a 3 mm depth in each ginseng taproot were made by using a sterile pipette tip with a 2 mm diameter. In this assay, added 10 μL of the NJ13 sterile filtrates (100% (v/v)) to wounds (each ginseng root has 3 to 5 wounds) in ginseng roots by a pipette. Fungal plugs of CBLJ-3 with a diameter of 8mm were cut at 1 cm from the edge of the colony incubated at 25 °C for 7 d on PDA medium, and were fastened to the surface of ginseng roots by sterile pipette tips.”
Line 188: “One day later, all of strain CBLJ-3’s hyphal plugs were removed.”
Line 191-192: “All experiments were repeated three times.”)
Point 10: Line 210: please, explain the PPI abbreviation.
Response 10: Thanks to the reviewer's suggestion, and we have explained the PPI abbreviation.
(Line 229: ”Protein-protein interaction (PPI)…”)
Results
Point 11: Line 239: "antibiological" is inappropriate word. Probably you mean antifungal?
Response 11: We mean that antifungal, and we have corrected the line 258 (“…antifungal…”).
Point 12: Line 241: antibacterial activity? Probably you mean "antifungal" or "antimicrobial"?
Response 12: We mean that antifungal activity, and we have corrected the line 260 (“…antifungal activity…”).
Point 13: Figure 1: "antagonistic" is a wrong term as you do not use co-cultivation of two strains, but use culture filtrate of one of tem. I consider "antifungal" would be better.
Response 13: Thanks to the reviewer's suggestion, and we have corrected the “Antifungal…” in the Figure 1.
Point 14: Line 249: the "co-culturing" term is used in the case of use of two live cultures. Since you use culture filtrate of NJ13, this term can not be used. Please, replace it with a simple "cultivation". Also, the agar medium used does not consist of only culture filtrate, so I suggest it would be beter to replace "consisting of" with "containing" or "supplemented with".
Response 14: Thanks to the reviewer's suggestion, and we have corrected the line 268 (“…cultivation… containing…”).
Point 15: Line 257: probably you mean "was measured" or "was determined" instead of "was treated"? Please, check.
Response 15: Thanks to the reviewer's suggestion. We have checked the sentence and also corrected the line 275-276 (“…was determined…”)
Point 16: Subsection 3.5: as I can see from the Fig. 5, there is also difference between the variants with different time of SM application; the variant when the SM was applied prior inoculation looks clearly better than that when it was applied after inoculation with the pathogen. I suggest it can be reflected in the text of this section.
Response 16: Based on valuable comments of the reviewer and the detailed analysis of the Figure 5, and the content of this section has been supplemented to the subsection 3.5.
(Line 307-310: “At last, there is also difference between ginseng roots with different time of the NJ13 sterile filtrates application, and ginseng roots when the NJ13 sterile filtrates were applied prior inoculation looks clearly better than that when NJ13 sterile filtrates were applied after inoculation with the I. robusta CBLJ-3.”)
Discussion
Point 17: Fig. 11: it is incorrect to use the "antibacterial molecules" term in relation to fungal cells. Please, check and correct.
Response 17: Thanks to the reviewer's suggestion, and we have corrected the “Secondary metabolites (SMs)” in the Figure 11.
Point 18: Line 541: I suggest "clarify" would be better than "illuminate".
Response 18: Thanks to the reviewer's suggestion, and we have corrected the line 572 (…clarify…)
